# Extracellular Vesicles from Mesenchymal Stem Cells as Potential Treatments for Osteoarthritis

**DOI:** 10.3390/cells10061287

**Published:** 2021-05-22

**Authors:** Nur Azira Mohd Noor, Asma Abdullah Nurul, Muhammad Rajaei Ahmad Mohd Zain, Wan Khairunnisaa Wan Nor Aduni, Maryam Azlan

**Affiliations:** 1School of Health Sciences, Health Campus, Universiti Sains Malaysia, Kubang Kerian 16150, Kelantan, Malaysia; nurazira@student.usm.my (N.A.M.N.); nurulasma@usm.my (A.A.N.); khairunnisaa94@student.usm.my (W.K.W.N.A.); 2Department of Orthopaedics, School of Medical Sciences, Health Campus, Universiti Sains Malaysia, Kubang Kerian 16150, Kelantan, Malaysia; rajaei@usm.my

**Keywords:** extracellular vesicles, exosomes, mesenchymal stem cells, osteoarthritis, chondrocytes

## Abstract

Osteoarthritis (OA) is a chronic degenerative disorder of the joint and its prevalence and severity is increasing owing to ageing of the population. Osteoarthritis is characterized by the degradation of articular cartilage and remodeling of the underlying bone. There is little understanding of the cellular and molecular processes involved in pathophysiology of OA. Currently the treatment for OA is limited to painkillers and anti-inflammatory drugs, which only treat the symptoms. Some patients may also undergo surgical procedures to replace the damaged joints. Extracellular vesicles (EV) play an important role in intercellular communications and their concentration is elevated in the joints of OA patients, although their mechanism is unclear. Extracellular vesicles are naturally released by cells and they carry their origin cell information to be delivered to target cells. On the other hand, mesenchymal stem cells (MSCs) are highly proliferative and have a great potential in cartilage regeneration. In this review, we provide an overview of the current OA treatments and their limitations. We also discuss the role of EV in OA pathophysiology. Finally, we highlight the therapeutic potential of MSC-derived EV in OA and their challenges.

## 1. Introduction

Osteoarthritis (OA) is a common degenerative disorder of the joints that affects the knee, hands, hip, spine and feet. Osteoarthritis represents a large and increasing health burden, causing patients’ function deterioration as well as extending public health costs to deal with an increasing OA prevalence worldwide. According to data from the United States Center for Disease Control and Prevention, OA affected 52.5 million people in the US in 2012 and is expected to affect 78 million people by 2040 [1]. In 2017, OA accounts for approximately 7.1% of the musculoskeletal disorders burden and showed a statistically significant increase in comparison to 31.4% in 2007 (95% CI:30.7,32.1) [2,3]. Meanwhile, between 1990 to 2019, the number of people affected globally by OA increased by 48% [4].

The prevalence of OA is increasing due to obesity and lifestyle. Patients with OA have limited daily activities and a greater risk of mortality [5]. Age is the main factor for OA due to ageing, muscle weakness and thinning of cartilage. Meanwhile, obesity is strongly associated with knee OA [6]. Apart from that, genetics and diet also influence the risk of OA. To date, the treatment for OA involves pain management using non-steroid anti-inflammatory drugs (NSAIDs) and pain killer to relive the symptoms [7] and surgical therapy for end-stage OA patients [8]. However, surgical therapy is costly and may lead to tissue hypertrophy. Therefore, appropriate therapeutic strategies are crucial to overcome existing problems. Recently, mesenchymal stem cells (MSCs)-derived extracellular vesicles (EV) therapy has been suggested as a potential therapeutic strategy for OA in an attempt to replace invasive treatments. MSC-derived EV is safe and a promising therapy due to their unique properties including a small size, stable culture, low immunogenicity, specific targeting and carrying biologically active components which make them a potential natural therapeutic delivery agent [9]. The small size of MSC-derived EV permits them to pass through cell barriers easily, thus increasing their capability in delivering genetic materials directly into the cytoplasm of the recipient cells [10,11].

This review highlights information regarding OA, MSCs and EV. In particular, this review focuses on the recent findings of potential therapeutic effects of EV derived from MSCs for OA treatment.

## 2. Pathophysiology of OA

Osteoarthritis is a progressive chronic condition that represents a pathological imbalance of degradative and reparative processes involving the entire joint and its component parts, with secondary inflammatory changes, particularly in the synovium as well as in the articular cartilage [12]. This complex process of disease involves biomechanical changes in joint composition, inflammation of the joints and metabolic changes which consequently lead to abnormal equilibrium of the synovial joint [13]. The definition and terminology of OA have long been a debatable subject centered on articular cartilage that loses its integrity [14]. Osteoarthritis is an active dynamic alteration arising from an imbalance between degradation and regeneration of diarthrodial joint tissues involving hyaline articular cartilage, subchondral bone, cruciate and collateral ligaments, capsule membrane and synovial membrane [15,16]. The key pathophysiological mechanisms in OA involve pro-inflammatory cytokines (interleukins (IL)-1, IL-6, IL-8 and tumor necrosis factor (TNF)-α) and pro-catabolic mediators through their signaling pathway and the well-characterized effect of nuclear factor κB (NFκB) and mitogen-activated protein kinase (MAPK) signaling responses and reprogramming are switching pathways in transcriptional networks (Figure 1) [17].

Cartilage consists of chondrocytes that produce a large amount of extracellular matrix. Chondrocytes in a healthy articular cartilage resist proliferation and differentiation, while chondrocytes in diseased cartilage proliferate and develop hypertrophy. The inflammatory mediators, mechanical and oxidative stress compromise the function and viability of chondrocytes, reprogramming them to undergo hypertrophic differentiation and early senescence, making them even more sensitive to the effects of pro-inflammatory and pro-catabolic mediators [18]. Products that are released from the cartilage matrix and chondrocytes in response to adverse mechanical forces and other factors induce the release of products that deregulate chondrocyte function via paracrine and autocrine mechanisms.

## 3. Treatment of OA

### 3.1. Osteoarthritis Management and Current Therapy

In view of the complexity of OA mechanism, a comprehensive plan for management of OA in different patients may include awareness in terms of educational plan, behavioral, psychosocial and physical interventions, as well as pharmacological treatment such as the use of topical, oral and intra-articular injection and also surgical treatment [19]. The principle of treatment should be addressed according to the degree of severity as well as the patient’s medical status to ensure that a personalized management strategy is tailored to their needs. The goals are to reduce symptoms and ultimately slow the disease progression, which may in turn improve quality of life and consequently reduce the health cost burden.

### 3.2. Pharmacological and Non-Pharmacological Therapy

A multidisciplinary, patient-centered combination of education, self-management, exercise, weight loss with realistic goals, encouragement and regular reassessment is recommended for individuals with OA [20]. Commonly, topical NSAIDs, oral NSAIDs and tramadol are used to treat OA. However, inconclusive evidence has been shown for acetaminophen, non-tramadol opioids and intra articular injections of corticosteroids, hyaluronic acid (HA) and platelets rich plasma (PRP) [21]. Clinical trials of intra articular glucocorticoid injections have demonstrated a short-term efficacy in knee OA as well as for hand OA [19]. However, a recent report raised the possibility that specific steroid preparations or a certain frequency of steroid injections may contribute to cartilage loss; however, the clinical significance of this finding was uncertain, particularly since change in cartilage thickness was not associated with a worsening in pain, function or other radiographic features [22].

Surgical intervention is indicated in end stage OA when all previous treatment methods have failed, a significant loss in quality of life and when the patient is severely symptomatic. Surgery is an effective treatment option; however, the long-term success rate is not clear and may fail depending on the longevity of the implant. Surgical techniques may include osteotomy around the knee, unicondylar arthroplasty (partial knee replacement) or total knee arthroplasty based on the patients’ condition and degree of deformity or joint destruction [23].

## 4. Mesenchymal Stem Cells

### 4.1. The Source of Mesenchymal Stem Cells, Isolation and Characterization

Mesenchymal stem cells (MSCs) were first discovered in 1966 from the bone marrow [24] and hold the concept from postnatal progenitor [25]. They are also known as multipotent mesenchymal stromal cells which possess two major features, including the ability to differentiate into multiple lineages and the capacity of self-renewal. According to the International Society for Cellular Therapy (ISCT), MSCs must meet three minimal criteria [26]. Firstly, MSCs must exhibit plastic-adherence when grown in vitro. Secondly, MSCs must express the surface antigens CD73, CD90 and CD105 while lacking expression of CD45, CD34, CD14 or CD11b, CD79α or CD19 and HLA-DR. Thirdly, MSCs must be able to differentiate into mesodermal cell types (i.e., adipocytes, chondrocytes and osteoblasts) when cultured under specific conditions.

Mesenchymal stem cells can be isolated from various tissues and are not restricted to mesodermal origin-type of cells such as bone marrow, adipose, muscle or bone. They were originally found in the bone marrow [24], but later studies identified MSCs in other tissues such as in the peripheral blood, brain, spleen, liver, kidney, lung, thymus, placental, umbilical cord and pancreas. Mesenchymal stem cells share a similar characteristic phenotype [26] despite their presence in different tissue sources, albeit with some additional features that represent their tissue origin [27]. Although MSCs can be isolated from almost every type of connective tissues [28], studies have shown that bone marrow represents the major source of MSCs [29,30,31].

### 4.2. Mesenchymal Stem Cell-Based Therapy

Mesenchymal stem cells have become a popular cell source for therapeutic purposes due to their immunomodulation and regenerative properties [32,33,34,35,36,37,38]. Mesenchymal stem cells possess the capacity to migrate to injured sites in response to other cells and environmental signals and also promote tissue regeneration orchestrated by the paracrine secretion of a broad repertoire of growth factors, chemokines and cytokines [39]. Through interaction with the host niche, MSCs were able to secrete bioactive mediators, such as growth factors, cytokines and EV that exert immunosuppressive, anti-apoptotic, anti-fibrotic, angiogenic and anti-inflammatory effects [40,41]. In addition, MSCs exhibited immunomodulatory functions by preventing immune cells’ activation or proliferation [42]. The immunomodulatory function of MSCs make them a good option for cell-based therapy, as the possibility for cell rejection is reduced.

The ability of MSCs to differentiate into various cell types enables their use as a tool in regenerative medicine. Mesenchymal stem cells have been reported to be successfully used as a therapy against many diseases and clinical conditions. At present, there are over 950 clinical trials worldwide that have used MSCs to treat various diseases [43], including bone and cartilage repair, diabetes, cardiovascular diseases, liver disease, immune-related, neurodegenerative diseases and spinal cord injuries. Mesenchymal stem cells have shown promising results in the clinical application of cell-based therapy. The efficacy of MSCs on bone regeneration in various orthopedic conditions has been widely demonstrated. Several clinical trials at different phases (I, II or III) have been performed for bone fracture repair using various sources of MSCs which were implanted either via direct injection or incorporated with osteogenic matrix or scaffolds which promoted bone repair and functions [44,45,46].

### 4.3. Mesenchymal Stem Cell-Based Therapy in Osteoarthritis Treatment

Regenerative medicine has become increasingly popular as a promising new approach for OA since articular cartilage has a limited capacity for spontaneous intrinsic repair, and may progress to OA, owing to the sparse distribution of highly differentiated chondrocytes, the low supply of progenitor cells and the lack of vascular supply [47]. This is evidenced by wide availability in clinical practice and by publications of many case series and clinical trials.

Cell-based therapy has been a promising option in OA because it is aimed at reversing the symptoms and pathophysiology of OA [48]. The ability of MSCs to differentiate between multiple lineages including musculoskeletal tissue supports the use of MSCs as an excellent source for degenerative musculoskeletal conditions such as in OA [49]. The therapeutic potential of MSCs in the treatment of OA is aimed at cartilage repair and restoration (Figure 2) [50].

The capability of MSCs to undergo chondrogenic differentiation assists in the regeneration of injured cartilage. In order to successfully enhance MSCs differentiation into chondrocytes, MSCs require soluble factors to promote differentiation such as transforming growth factor (TGF)-β, bone morphogenetic protein (BMP) and insulin growth factor (IGF)-1 [51,52]. The combination of BPM2/TGF-β in pre-chondrogenic medium resulted in high expression of chondrocytes specific genes such as collagen type II alpha 1 (COL2A1) and aggrecan (ACAN) [53]. Mesenchymal stem cells proliferation could also be enhanced by a low oxygen condition, which results in an increase of cartilage specific genes expression (COL2A1 and SOX9) and proteoglycan synthesis during chondrogenic differentiation [54]. Besides being able to regenerate or restore injured cartilage, MSC-based therapy also focuses on the inflammation attenuation. MSCs were shown to orchestrate immunomodulatory function of inflammatory responses through paracrine activities [55]. Adipose tissue-derived MSCs co-cultured with chondrocytes or synoviocytes showed a decrease in the expression of inflammatory factors such as IL-1β, TNF-α, IL-6 and CXCL8/IL-8 through the COX-2/PGE2 pathway as modulators, exerting anti-inflammatory effects in OA condition [56].

There were several strategies used by manipulating different MSCs sources and conditions for potential OA treatment [57]. Different sources of MSCs contribute to different outcomes in cartilage regeneration. For example, human skeletal stem cells (hSSCs) have shown the ability to generate into multilineage ossicles containing bone, cartilage and stroma, which have a potential for chondrogenic and osteogenic activities [58]. Apart from selecting different sources of MSCs, manipulating the MSCs in an in vitro culture condition also enhances chondrogenesis. A co-culture of human chondrocytes and chondro-progenitors (also known as cartilage cells) resulted in increased chondrogenic ability and increased cytokines and growth factors compared to bone-marrow derived MSCs [59]. Furthermore, controlling the culture condition also contributes to better chondrogenic differentiation such as culturing MSCs with TGF-β [60] or fibroblast growth factor (FGF)-2, 9 and 18 [61].

Direct intra articular injection of MSCs has been shown to improve the condition of OA patients [62]. Several studies have reported the use of intra articular injection of autologous bone-marrow-derived MSCs, which showed improvement in clinical symptoms and resulted in higher arthroscopic and histological grades than the control group [63,64]. In another study, an intra articular injection of autologous MSCs showed significant growth of cartilage and meniscus on magnetic resonance imaging (MRI), increased range of motion and modified visual analog scale (VAS) pain scores at 24-weeks post injection [65]. Intra articular injection of autologous bone marrow-derived MSCs showed significant improvement in terms of knee pain and quality of life during the 6-month follow-up [66].

Articular cartilage tissue engineering is another aspect that can be manipulated by using biomaterial constructs. Several biomaterials such as fibrin [67], biopolymer (chitosan) [68], synthetic polymers [69] and hydrogel [70] were widely used for their ability to degrade rapidly, their low immunogenicity and their ability to enhance MSCs proliferation and chondrogenic differentiation.

Another strategy to achieve chondrogenic differentiation of MSCs is via extracellular matrix (ECM) application by using synthetic or natural scaffolds. MSCs cultured in vitro in serum free conditions or in hydrogel or scaffold materials such as polymers, alginate beads and collagen sponges will eventually promote the differentiation of MSCs into chondrocytes [71]. Early pre-clinical studies have showed promising chondrogenic differentiation properties when MSCs were seeded on scaffolds [72,73]. Autologous transplantation of rabbit MSCs with HA gel sponges showed a well-repaired cartilage tissue which resembles the articular cartilage of the surrounding structure [73]. Another study has also used intra-articular injection of autologous adipose MSCs combined with HA which suppressed the OA progression and promoted cartilage regeneration [74]. The first autologous bone marrow-derived MSCs transplant embedded in vitro in a collagen gel was reported in 2004. A year later, the use of the matrix-induced autologous chondrocyte implantation (MACI) technique was reported by using a collagen bilayer seeded with chondrocytes [75]. Since then, many studies reported promising findings by using the MACI technique with various types of scaffolds, especially collagen due to its natural presence in the ECM [76]. Articular cartilage regeneration using allogeneic human umbilical cord blood-derived MSCs which were expanded in HA hydrogel demonstrated production of hyaline-like cartilage after one year and the regenerated cartilage persisted after three years [77].

The use of MSCs for articular cartilage regeneration has several limitations. A transcriptome analyses of human neonatal articular cartilage (hNAC) and MSC-derived cartilage has demonstrated that over 500 genes that are highly expressed in hNAC were not expressed in MSC chondrogenesis [78]. This suggests that in vitro MSC-derived cartilage may not represent the actual in vivo chondrocytes. In addition, in vitro cultured MSCs displayed a hypertrophic phenotype, thus limiting their application in articular cartilage tissue engineering [79]. Since MSCs are heterogenous, their numbers vary in tissues and different MSC sources may not be equivalent in terms of function [79,80].

## 5. Extracellular Vesicles

### 5.1. Extracellular Vesicles Biogenesis

Extracellular vesicles consist of exosomes and microparticles (MP) [81]. The biogenesis of EV varies depending on their subtypes (Figure 3). The generation of exosomes involves three main stages, which are endosomes, multivesicular bodies (MVBs) and exosomes [82].

During the endosomes stage, the formation of early endosomes is characterized by a tube-like shape, which occurs when endocytic vessels are transferred and located closer to outer edge of the cytoplasm. Subsequently, early endosomes further mature into spherical late endosomes and are located near the nucleus. The second stage of exosomes formation involves degradation of late endosomes (also known as MVBs) that carry intraluminal vesicles upon fusion with lysosomes [83]. Exosomes are subsequently released from MVBs into the extracellular space through exocytosis of plasma membrane [84]. In contrast, MP are generated from outward blebbing of the plasma membrane through two main steps, which are the rearrangement of cytoskeleton and externalization of phosphatidylserine (PS) [85]. In response to cell activation, an increase of intracellular calcium activates calcium-dependent enzymes such as kinase, calpain and gelsolin and also inhibits phosphatase [86] by facilitating the cleavage of cytoskeleton proteins [87]. Calcium influx also leads to the activation of cytosolic enzymes such as flippase, floppase and scramblase [88] that mediate the externalization of PS and phosphatidylethanolamine, while internalizing phosphatidylcholine and sphingomyelin [89]. Thus, cytoskeleton proteolysis and phospholipid imbalance favor the cellular blebbing, which ultimately leads to the shedding of MP [90]. In OA patients, EV may be generated from articular cartilage, which is known as articular cartilage vesicles due to the pathological process of OA [91], while EV in the synovial fluid is possibly released from chondrocytes and synoviocytes [92].

### 5.2. Isolation and Characterization of Extracellular Vesicles

The choice of EV isolation method is crucial, as contaminants may attribute to alteration of EV function [93]. Isolation of EV may be performed by using several protocols including centrifugation, precipitation and chromatography (Table 1).

Differential centrifugation followed by ultracentrifugation is the most widely used protocol to isolate EV [102] from OA patients. This technique allows the removal of cell from the synovial fluids, since slow initial centrifugation results in cell precipitation. Meanwhile, ultracentrifugation at 20,000 × *g* permits the recovery of MP without exosomes contamination [103], while exosomes can be isolated from the synovial fluid with ultracentrifugation at 100,000 to 200,000× *g* [104]. However, the ultracentrifugation method requires a large volume of synovial fluid, is time-consuming and is only effective for body fluids with low viscosity [105]. Extracellular vesicles can also be isolated using the precipitation method, in which synovial fluid is added into large polymers such as polythrilenglycol and polyethylene glycol (PEG), followed by EV precipitation [105,106]. Although the concentration of purified EV using the precipitation method was high, the purity and accuracy in terms of size distribution of EV was low [107]. It has been shown that EV isolated by this method exerted similar effects in an ischemic stroke model and their corresponding cells [106,108]. Additionally, size-exclusion chromatography isolates EV from synovial fluid precisely based on size without affecting the structure of EV [105]. In fact, size-exclusion chromatography involves the penetration of particles through a column, where larger particles will be eluted prior to the smaller particles [109].

The use of single isolation method usually results in low specificity; therefore, the combination of techniques in order to achieve better specificity of EV isolation is recommended. Apart from the techniques discussed, other techniques such as ultrafiltration, washing with EV-free buffer [110] and two-step centrifugation are commonly used. In addition, isolation of EV using combination methods of ultracentrifugation with ExoQuick^TM^ precipitation resulted in high recovery of EV, while combination of ultracentrifugation with density gradient centrifugation permits isolation of EV with intact morphology [111].

Several techniques have been widely implemented for EV characterization based on physical as well as chemical, biological and compositional analysis (Table 2).

Physical analysis of EV is commonly performed by using electron microscopy, including scanning electron microscopy (SEM) [112] and transmission electron microscopy (TEM). The use of electron microscopy assists in the visualization of EV morphology at a high resolution image [109]. However, this technique limits multi-parametric phenotypic EV characterization, and lengthy sample preparation. Apart from that, nanoparticle tracking (NTA) is useful in EV characterization, as this technique provides qualitative analysis in terms of size and concentration of EV [109,113]. NTA also allows appropriate resolution in characterizing individual particles of EV. Another technique that allows physical characterization of EV is dynamic light scattering (DLS). In contrast to EM, DLS provides information regarding the average of the size distribution of EV by determining collective mobility of EV instead of a single EV [114,115].

Furthermore, characterization of EV could be performed by implementing chemical, biological and compositional analysis. Flow cytometry (FCM) is the most widely used technique to characterize EV. This method permits the analysis of large number of samples in a short time [88] and also provides high resolution of quantitative and qualitative data based on individual EV. Currently, the characterization of EV is ascertained by Western blotting, which mainly assesses the markers expressed on EV; thus, this technique appears as a conformational technique of EV [116]. Western blotting also enables the detection of both surface proteins and internal proteins of EV [109]. However, the use of Western blotting is limited in translational studies, as it requires EV in a large quantity [117]; as additionally, the specificity and reproducibility of EV analysis may interfere with the quality of the antibodies used [109].

### 5.3. Extracellular Vesicles in Osteoarthritis

Cells in tissue and leukocytes that infiltrated the joints affected with arthritis may release EV into the extracellular space such as in the synovial fluid. It has been previously reported that elevated levels of EV in synovial fluid from OA patients were capable of triggering synoviocytes to secrete cytokines and chemokines [118]; thus, EV could potentially act as a biomarker in OA. Previous study has shown the upregulation of microRNA (miR)-16-2-3p and downregulation of miR-26a-5p, miR-146a-5p and miR-6821-5p in synovial fluid-derived EV from female patients with OA compared to non-OA female patients [119]. In contrast, down regulation of miR-6878-3p and upregulation of miR-210-5p were observed in synovial fluid-derived EV from male OA patients compared to non-OA male patients. These results suggest that EV may be used as potential OA biomarkers in a gender-specific manner. However, further studies at the molecular level are necessary for a better understanding of EV. The use of EV as potential biomarkers in other arthritis-related diseases, particularly RA, has been previously reported. For instance, a significant increase of four-fold of Hotair expression was reported in exosomes from RA patients compared to non-RA patients [120]. Hotair is a long non coding RNA (IncRNA) that modulates the migration of active macrophages to the site of inflammation. The expression of Hotair was downregulated in EV derived from blood of a non-RA patient with a high C-reactive protein (CRP). This suggests that EV may be used as a potential biomarker to diagnose RA [120].

As EV can be released by various cell types, EV also carry genetic and cytosolic components which are similar to their origin cells including cytosolic proteins, mRNA, miRNA and small non-coding RNAs such as long non-coding RNA (lncRNA) and circular RNA (circRNA) [121]. These cargos of EV modulate gene expression, altering the downstream functions and behavior of recipient cells as well as exerting physiological and pathological effects [122]. In healthy individuals, EV derived from chondroblasts and osteoblasts in the developmental phase were actively involved in the process of chondrogenesis by accumulating calcium and inorganic phosphate from ECM, which results in mineralization in the lumen and subsequently form hydroxyapatite crystals [91]. Several proteins and growth factors such as BMP and vascular endothelial growth factor (VEGF) have been reported to exist in EV [123]. This finding indicates that EV are also involved in angiogenesis, a process of blood vessel formation as well as in chondrocytes and osteoblast differentiation in growth plate. In addition, EV derived from normal human AC or AC vesicles were responsible for the neutralization of adenosine triphosphate (ATP), calcium and inhibition of phosphorylation, which may be deleterious to adjacent chondrocytes [91].

To date, the role of EV in the pathogenesis of OA has been poorly understood. However, previous study has reported that OA pathogenesis may be driven by interaction between resident cells and immune cells, ECM of various tissues and also the synovial fluid [124]. Meanwhile, EV derived from cells within the joints may mediate the pathogenesis and progression of OA by assisting these cell-cell communications [125]. Previous study has suggested that in OA pathogenesis, EV mediates the activation of fibroblast-like synoviocytes by synovial macrophages and infiltrating leukocytes in the synovial membrane [92]. In response to activation, synoviocytes further release cytokines and enzymes, thus retaining joint inflammation. Additionally, EV may promote changes in subchondral bone and matrix degradation. It has been demonstrated that EV derived from chondrocytes of OA patients engage with secretion of atypical protein and enable the transfer of information between cells and pathological calcification in articular cartilage [126]. Treatment of EV from OA on macrophages resulted in secretion of proinflammatory cytokines and chemokines such as matrix metalloproteinases (MMP)-7, MMP-12, IL-1β, CXCL1, CCL8, CCL15 and CCL20, which initiate cartilage degradation and inflammation in the joints [107]. In articular chondrocytes, treatment of synovial fluid-derived EV from OA decreases cell survival and downregulates anabolic genes expression including COL2A1 and ACAN, while increases catabolic and inflammatory gene expression including IL-6 and TNF-α [119]. The role of EV in communication between fibroblast-like synoviocytes and chondrocytes was previously investigated. It was found that exosomes derived from IL-1β-stimulated synovial fibroblasts resulted in OA-related gene expression in articular chondrocytes such as ACAN and MMP-13 [127], which further leads to degradation of ECM, thus promoting the progression of OA.

### 5.4. Therapeutic Potential of MSC-Drived EV in Osteoarthritis

The capability of cartilage to regenerate or self-recover in OA is limited. Mesenchymal stem cells-derived EV offer a new therapeutic strategy for OA. The mechanism regulated by MSC-derived EV in OA treatment may be mainly through their genetic components that promoting paracrine action [128,129]. Mesenchymal stem cells-derived EV also possess immunomodulatory properties. A previous report demonstrated that MSC-derived EV were responsible for suppressing pro-inflammatory cytokines and elevating anti-inflammatory cytokine secretions [130]. This suggests that MSC-derived EV could potentially be used in future OA treatment.

The potential of MSC-derived EV in OA treatment has been extensively studied (Table 3). A previous study has demonstrated that MSC-derived EV protected a collagenase-induced OA mice model from joint damage [131]. Such protection includes prevention of bone and cartilage in OA, which could be due to upregulation of COL2A1 and ACAN, downregulation of MMP-13, a disintegrin and metalloproteinase with thrombospondin motifs (ADAMTS)-5 and inflammatory markers, as well as preventing apoptosis of chondrocytes and macrophage activation. Additionally, the injection of intra articular of EV derived from human embryonic MSCs in femur resulted in regeneration and repair of osteochondral defect, subchondral bone and cartilage, as well as deposition of normal ECM in a rat OA model [132]. A comparativestudy between synovial membrane MSC-derived EV (SMMSC-EV) and EV secreted from induced-pluripotent stem cell-derived MSCs (iMSC-EV) found that both SMMSC-EV and iMSC-EV treatment stimulated the proliferation and migration of chondrocytes and attenuated OA in a collagenase-induced mouse model [133]. Similarly, treatment with EV from human bone marrow-derived MSCs (hBMSC-EV) promoted cartilage repair by triggering the production of ECM by OA chondrocytes and improving inflammatory response in vitro [134].

Osteoarthritis treatment with EV from adipose tissue-derived MSCs has shown a promising alternative therapy. It was demonstrated that infrapatellar pat pad MSC-derived EV may prevent cartilage degradation and enhance the chondrocytes autophagy level by inhibiting the rapamycin signaling pathway [136].

Since heterogenous cell entities of MSCs may give rise to different EVs with different functions, whether they exert similar therapeutic effects on OA is still questionable. In addition, different EV preparations may also influence their therapeutic activities. Furthermore, not all MSC sub-types are capable of mediating clinical impacts on OA, and neither are their EVs. Therefore, further study on MSC-derived EVs is needed for better understanding of their mechanism in OA treatments.

### 5.5. The Promise and Challenges of EV as a Therapeutic Delivery System

Recently, MSC-derived EVs have obtained interest from regenerative medicine due to their high therapeutic efficiency. They have been suggested as a clinical effective delivery agent in OA treatment, as they meet the focus of clinical therapy by exerting two important therapeutic effects including cartilage protection and regeneration as well as an anti-inflammatory effect. Additionally, MSC-derived EVs is a promising therapeutic agent with high sustainability, a non-invasive collection process and highly reproducible and safe characteristics [137], which is indicated by its low toxicity and low immunogenicity [138]. The use of MSC-derived EVs also confers a few other benefits such as the ability to cross the biological barriers [139], as they are able to communicate directly with the target cells, thus offering rapid clearance, lowering the risks, and reducing toxicity [140]. Moreover, MSC-derived EVs may avoid immunogenic reactions such as immune rejection due to a lack of MHC class I/II [141].

Despite the therapeutic potential of MSC-derived EVs, a number of challenges remain, including exosome molecular diversity, a lack of exosomal targeting properties [142], and excessive transfer of gene information [143]. A large-scale MSC-EV isolation process, for example, may pose technical and experimental challenges. The existing EV generation methods such as tissue culture methods in flask limits the large-scale production of EV, thus hindering the use of EV in clinical therapeutic applications [144]. Although sustainable quantities of EV may be harvested through the long-term passaging method, this method may lead to differentiation of MSC [145]. Apart from that, reliable EV characterization methods, rapid and precise methods for characterizing EV functional cargo, as well as the pharmacokinetics and transfer mechanism of MSC-EVs remains unclear. MSC-EV has been suggested to have a therapeutic effect via miRNA, but cytoplasmic and membrane proteins, mRNA and small non-coding RNA, all of which can be passed to recipient cells [146] but are not always evaluated.

## 6. Conclusions

In conclusion, both MSCs and EVs play a significant role in immunomodulation of OA. Therefore, MSC-derived EV-based therapy has a potential role in enhancing damaged articular tissue repair. As MSC-derived EV are secreted under physiological and pathological conditions, they may exert different effects via different pathways. The membrane as well as cytosolic protein and lipids and genetic components including mRNAs siRNA, miRNAs and ribosomal RNAs of EV are the main factors that allow them to facilitate cell-to-cell interaction. MSC-derived EV therapy offers a potential safe approach for OA treatment. Studying the potency assay of MSC-derived EVs is challenging. 

Although previous findings have demonstrated a promising therapeutic effect of MSC-derived EVs which was similar to MSC; however, several critical issues should be considered in designing MSC-derived EV therapy, including the source of EV as distinct MSC may give rise to different EV subtypes with different biological effects and standard EV preparation protocols, since independent MSC-derived EVs preparation may differ in regards to their therapeutic potentials, a safe and effective administration route with proper targeted treatment sites as well as a sufficient dose and frequency of EV administration to inflict an optimal therapeutic effect on OA or arthritis-related disease generally. Therefore, further functional testing of MSC-derived EVs, particularly for OA treatment, is necessary.

## Figures and Tables

**Figure 1 cells-10-01287-f001:**
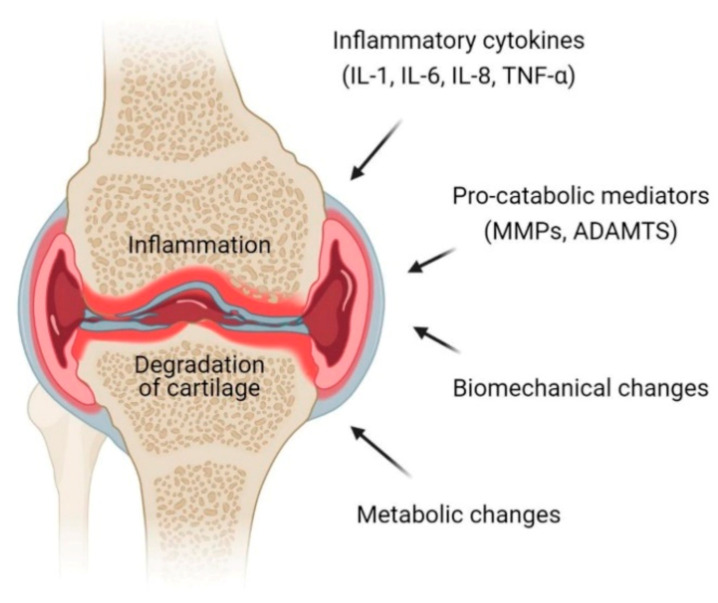
Pathophysiology of osteoarthritis (OA) at the knee joint. Inflammation and degradation of cartilage at the joint are common features of OA which resulted from the release of proinflammatory cytokines including interleukin (IL)-1, IL-6, IL-8 and TNF-α, pro-catabolic mediators such as matrix metalloproteinases (MMPs) and a disintegrin and metalloproteinase with thrombospondin motifs (ADAMTS), biomechanical changes and metabolic changes.

**Figure 2 cells-10-01287-f002:**
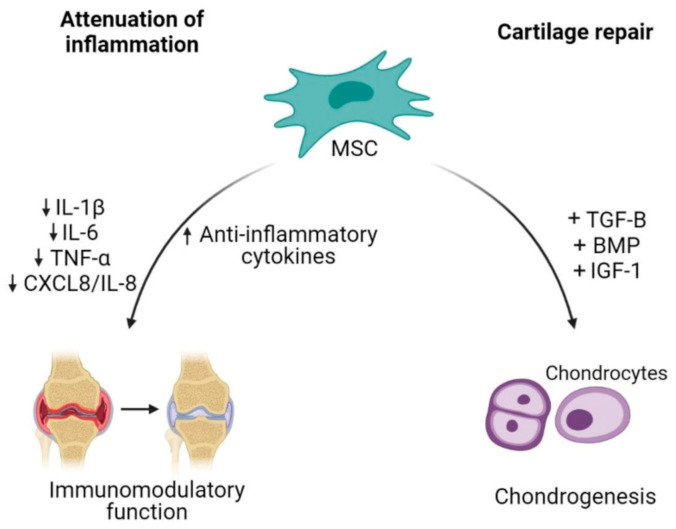
Mesenchymal stem cells (MSCs) as an alternative cell-based therapy for osteoarthritis. Mesenchymal stem cells dampen inflammatory activities in OA by reducing the releases of pro-inflammatory cytokines and increasing anti-inflammatory cytokines. Additionally, their plasticity characteristic allows differentiation of MSCs into chondrocytes, thus contributing to cartilage repair.

**Figure 3 cells-10-01287-f003:**
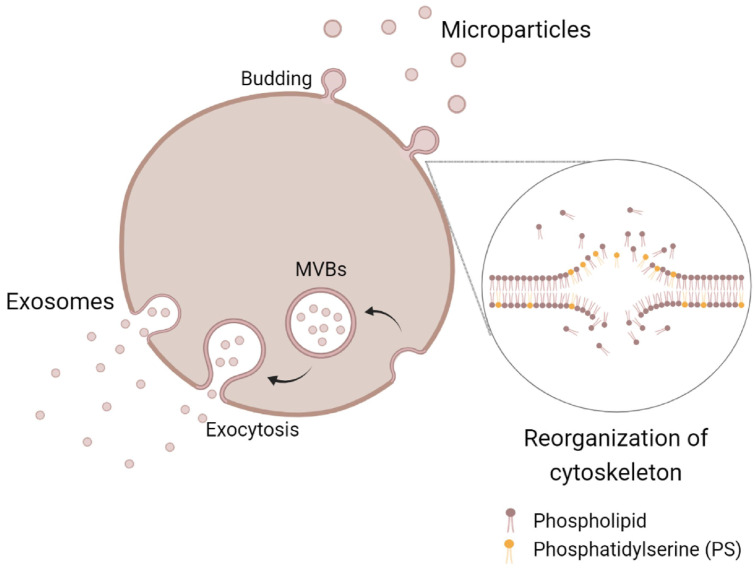
The formation of extracellular vesicles (EV). Exosmes are released from multivesicular bodies (MBVs) through exocytosis, while microparticles (MP) shed from cell membrane through budding.

**Table 1 cells-10-01287-t001:** Advantages and disadvantages of EV isolation methods.

Isolation Method	Advantages	Disadvantages	Yield
Differential centrifugation/ultracentrifugation	EV subtypes isolation [94], cost effective	Time consuming, less effective for body fluid with high viscosity, low purity	Intermediate
Precipitation	High EV recovery	Low specificity, less accurate in terms of size distribution, and low purity	High
Size exclusion chromatography	Precise, structurally unaffected of EV	Quantitatively inefficient, and time consuming	Intermediate
Ultrafiltration	EV subtypes isolation based on size [93], and cost effective	Low specificity, and time consuming	Low [95]
Field-flow fractionation (FFF)	High specificity [96], accurate EV size distribution and High EV integrity	Small volume of sample	Intermediate
Commercial kits (eg: ExoQuick, ExoMir kit)	High EV integrity, convenient procedure [97]	Costly, low purity and low reproducibility	Intermediate
Immunoprecipitation	High purity [98], EV subtypes isolation based on protein marker [99]	Costly and time consuming [100]	Intermediate
Immunoaffinity columns	Fast and high reproducibility	Low specificity [101]	Intermediate

**Table 2 cells-10-01287-t002:** Different techniques for EV characterization.

	Techniques	Advantages	Disadvantages
Physical	Electron microscopy (SEM and TEM)	Allow assessment of EV morphology	Time consuming, single parametric phenotypic EV characterization
	Nanoparticle tracking (NTA)	Allows assessment of individual EV in terms of size and concentration of EV	Starting amount of EV and contaminants may affect the accuracy of results
	Dynamic light scattering (DLS)	Fast, small starting amount of EV, provides size range of EV	Limits the analysis of individual EV
Chemical, biological and compositional analysis	Flow cytometry (FCM)	Quantitative and qualitative analysis, EV-subtypes analysis, permits analysis of large numbers of samples at a time	Occurrence of swarm detection, overlapping background noise and minimal detection limits
	Western blotting	Assess markers of EV as well as internal proteins of EV	Limited in translational studies, the quality of antibodies used may compromise the specificity of the analysis

**Table 3 cells-10-01287-t003:** Therapeutic evidence of MSC-derived EV in OA treatment.

Type of EV	Model	Marker	Time Point of Assay	Specific Characteristic of In Vivo or In Vitro Studies	Findings	References
** Exosome derived from human bone marrow-derived MSCs (MCS-Exos and MSC-miR92a-3p-Exos) **	Human bone marrow MSCs (normal and OA) Mouse (collagenese-induced OA)	MSCs: CD73, CD90, CD105 MSC-ExoS: CD9, CD63, CD81, and HSP70	**Proliferation****assay.** MCS-Exos and MSC-miR92a-3p-Exos were incubated with normal and OA chondrocytes for 0–5 days **Transfection.** MSCs were transfected with miR-92a-3p mimic or inhibitor for 48 h (qRT-PCR) and 72 h (western blot) **In vivo study.** MCS-Exos and MSC-miR92a-3p-Exos were injected into collagenase-induced OA mice after 7, 14 and 21 days following OA induction	**Proliferation****assay.** 200 μg exosomes/mL were used **Transfection.** 50 nM of miR-92a-3p mimic or inhibitor were used **In vivo study.** 500 μg/mL of MSC-Exos and MSC-miR-92a-3p-Exos were used	Both MSC-Exos and MSC-miR92a-3p-Exos significantly enhance chondocytes proliferation compared to control group, where MSC-miR92a-3p-Exos exert a more potent effect compared to MSC-Exos. MSC-miR92a-3p-Exos significantly upregulated ACAN, COL2A1, SOX9, while COL10A1, RUNX2, MMP-13 and WNT5A were significantly downregulated. This indicates the capability of MSC-miR92a-3p-Exos in enhancing cartilage development. Both MSC-miR-92a-3p-Exos and MSC-Exos prevented cartilage matrix loss compared to the OA group, where the level of COL2A1 and ACAN were significantly better with the presence of MSC-miR-92a-3p-Exos.	[135]
**Exosomes and microparticles (MP) derived from murine bone marrow-derived MSCs**	Chondocytes from C57BL/6 mice (IL-1β-induced OA-like phenotype) Mouse (collagenese-induced OA)	MP: CD29, CD44 and Sca-1 Exosomes: CD9, CD81	**Cartilage restoration.** MP and exosomes were incubated with chondrocytes following OA induction for 24h **Apoptosis induction****.** MPs or exosomes were added into chondrocytes-MSCs cocultures for 6 h **Monocytes activation.** MP and exosomes were incubated with murine spleen-derived macrophage for 3 days **In vivo study**. MPs and exosomes were injected into mouse at day 7 after OA induction and harvested at day 42	**Cartilage restoration.** 12.5 ng; 125 ng or 1.25 µg of MP and exosomes were used **Apoptosis induction.** 125 ng or 250 ng of MP and exosomes were used **Monocytes activation.** 50 ng of MPs or exos were used **In vivo study.** 500 ng/5 µL of MP or 250 ng/5 µL of exosomes were used	Both MPs and exosomes significantly upregulated ACAN, COL1 and COL2B expression in a dose-dependent manner and down-regulated MMP-13, ADAMTS-5 and inflammatory iNOS, thus indicating MP and exosomes exhibited a condroprotective effect MPs and exosomes prevented apoptosis and reduced the level of apoptotic chondrocytes with significantly lower doses than MSCs, where Exos exert a more potent effect compared to MPs. MP and exosomes inhibiedt the expression of CD86, MHCII or CD40 as well as downregulated TNF-α and upregulated IL-10, and thus possessed an immunosuppresive effect MP and exosomes significantly improved volume, cartilage degradation (surface/volume ratio) and thickness of articular cartilage, indicating protection of cartilage degradation	[131]
**Exosomes derived from human embryonic stem cell (HuECS)-derived MSCs**	Rat (osteochondral defect)	Exosomes: CD81, TSG101	Intra-articular injections of exosomes or PBS were weekly administered at the site of osteochondral defect for 12 weeks and harvested at weeks 6 and 12	100 μg exosomes was administered	At 12 weeks, exosomes promoted almost complete neotissue coverage with good surface regularity and complete integration with the adjacent cartilage. Histologically, exosomes enhanced the formation of hyaline cartilage indicated by uniform and intense staining of GAG (>80%), high level of type II collagen, and low level of type I collagen. Meanwhile, almost all cells in the repaired tissue injected with exosomes appeared to be chondrocytic, and displayed pericellular matrix staining of type VI collagen. It also showed good integration with adjacent cartilage and subchondral bone.	[132]
**Exosomes derived from human synovial membrane-derived MSCs (SMMSC-Exos) and induced pluripotent stem cell-derived MSCs (iMSC-Exos)**	Mouse (collagenase-induced OA)	iMSCs: CD29, CD44, CD73 and CD90 SMMSCs: CD44, CD73, CD90 and CD166 iMSC-Exos and SMMSC-Exos: CD9, CD63, and TSG101 proteins	**In vivo study.** iMSC-Exos SMMSC-Exos were injected intra-articularly at knee joints of normal and OA mice on days 7, 14 and 21 following OA induction **Chondrocytes migration assay.** iMSC-Exos or SMMSC were incubated with scratched monoclayer chondrocytes, and wound closure was monitored at 0 h, 24 h and 48 h **Chondrocytes prolifertaion assay.** iMSC-Exos or SMMSC-Exos were incubated with chondrocytes for 5 days	**In vivo study.** iMSC-Exos and SMMSC-Exos in PBS at concentration of 1.0 × 10^10^/mL were used **Chondrocytes migration assay.** chondrocytes were cultured in DMEM F-12 medium containing 10^8^/ml iMSC-Exos or 10^8^/ml SMMSC-Exos **Chondrocytes prolifertaion assay.** Chondrocytes were seeded at 2 × 10^3^ cells/well and cultured for 8h before were incubated with 10^7^ exosomes/ml or 10^8^ exosomes/ml were added	iMSC-Exos enhance hyaline formation features with a smooth cartilage surface, regular cellular organization and normal proteoglycan content, while SMMSC-Exos showed a moderate surface irregularity and superficial fibrillation compared to OA group. Meanwhile, iMSC-Exos were more effective in inhibiting loss of proteoglycan in cartilage compared to SMMSC-Exos indicated by a reduction in safranin O staining in the SMMSC-Exos group compared to the iMSC-Exos group Both iMSC-Exos and SMMSC-Exos significantly enhanced chondrocytes motility compared to OA group, with iMSC-Exos significantly increased chondrocytes motility at 24 h and 48 h compared to SMMSC-Exos iMSC-Exos and SMMSC-Exos stimulated chondrocyte proliferation at 10^8^ exosomes/ml compared to the control group, while iMSC-Exos exhibited a more potent effect	[133]
** EVs secreted from human bone marrow-derived MSCs (BMMSC-Evs) **	Human knee cartilage of OA patient (TNF- α -induced inflammatory)	BMMSC-Evs: CD9 and CD63	**Inflammation inhibition assay.** BMMSC-Evs were incubated with TNF-α treated OA chondrocytes for 48 h **Collagenase activity assay.** BMMSC-Evs were incubated with TNF-α treated OA chondrocytes for 4 h **Proliferation assay.** BMMSC-Evs were incubated with TNF-α treated OA chondrocytes for 5 days in the presence of 5-ethylnyl-2’-deoxyuridine (Edu) **Cartilage regeneration.** cells were cultured in fibrin constructs and treated with BMMSC-Evs every 5 days for 28 days	** Collagenase activity assay. ** BMMSC-Evs were incubated with OA chondrocytes at an ambient temperature, protected from light **Proliferation assay.** BMMSC-Evs were incubated with OA chondrocytes in the presence of 10 μM 5-ethylnyl-2’-deoxyuridine.	BMMSC-EVs significantly downregulated the expression of TNF-alpha-induced COX2, IL-1 alpha, IL-1 beta, IL-6, IL-8 and IL-17 in OA chondrocytes, which suggests anti-inflammatory potential of BMMSC-Evs by inhibiting phosphorylation of IκBα. BMMSC-Evs inhibited increasing collagenase activity induced by TNF-α, and also promoted OA chondrocytes proliferation and revoked the inhibitory effect of TNF-alpha BMMSC-EVs significantly promoted the production of proteoglycans in the newly formed tissue, induced the expression of ACAN, COL2A1, SOX9 and WNT7A, downregulated RUNX2 and COL10A1 as well as enhanced type II collagen production in OA chondrocytes.	[134]

## Data Availability

No new data were created or analyzed in this study. Data sharing is not applicable to this article.

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
