# Peer review of "Extracellular Vesicles from Mesenchymal Stem Cells as Potential Treatments for Osteoarthritis"

_cells, 2021, doi:10.3390/cells10061287_

Round 1

Reviewer 1 Report

This is well written review article. I feel it contains important information to the readers, since there is no such review focusing on the extracellular vesicles from the MSCs.

Reviewer 2 Report

General comments:

This is a well-written review, but in my opinion and based on the topic indicated in the title, the space of the manuscript devoted to specific contents should be revised.

Too much space was devoted to the general part relating to the pathophysiology/management of osteoarthritis (3 pages) and mesenchymal stromal cells (3 pages), to the detriment of the extracellular vesicles (about 4 pages) which should be the protagonists.

In this regard, I suggest the authors to shorten the more general parts, focusing on cell-based or cell-free injective treatments for the management of OA, along with on the role of the mesenchymal stromal cells on OA.

Moreover, as you stated in conclusions “study on potency assay of MSC-derived EV is challenging”, I suggest to add (or replace the more general figures) a table with the specific characteristics of the in vitro and in vivo studies performed using EVs for the OA treatment and specifying the model, the markers, the kind of EVs, the time-point of the assays used in the different papers, to give the readers an useful overview of the state of the art.

Specific comments:

-Introduction, lines 47-49: this sentence is too strong. No clinical trials sustaining this statement are reported. Please, use “proposed” and change the last part of the sentence i.e. “in the attempt to replace…”.

-Page 4, 3.4 sub-section: too much space is devoted to ACI/MACI techniques. At this purpose, please consider that these techniques are used for the treatment of knee cartilage defect and not in the context of a diffuse OA. In this paragraph should be discussed also the MSC treatment. I suggest shortening the general and well-consolidated part on MSCs, focusing on their role in OA. Please, insert in the bibliography and use information from this review: “Cytotherapy. 2019 Dec;21(12):1179-1197. doi: 10.1016/j.jcyt.2019.10.004. Mesenchymal stem cells in the treatment of articular cartilage degeneration: New biological insights for an old-timer cell”, where an extensive discussion of these topics is reported.

-Page 6: 4.3 sub-section: please, insert in the bibliography and use information from this review: “Knee Surg Sports Traumatol Arthrosc. 2019 Jun;27(6):2003-2020. doi: 10.1007/s00167-018-5118-9. Injective mesenchymal stem cell-based treatments for knee osteoarthritis: from mechanisms of action to current clinical evidences” to add a more clinical message.

-Page 14, lines 565-579: this part seems to be quite disconnected from the rest of the paragraph. If you want to maintain some information contained in this part, please move it in a more general part on EVs.

Round 2

Reviewer 2 Report

I am satisfied with the changes made by the Authors.
I thank the Authors for making the review more focused on the subject and for providing more detailed and timely useful information to readers about the studies on the subject.
I would suggest to break up the nine-line sentence added in the conclusions for greater readability.  

This manuscript is a resubmission of an earlier submission. The following is a list of the peer review reports and author responses from that submission.